# Silent synapses generate sparse and orthogonal action potential firing in adult-born hippocampal granule cells

Liyi Li[1], Sébastien Sultan[2], Stefanie Heigele[1], Charlotte Schmidt-Salzmann[3], Nicolas Toni[2], Josef Bischofberger[1]*

[1]Department of Biomedicine, University of Basel, Basel, Switzerland; [2]Department of Fundamental Neurosciences, University of Lausanne, Lausanne, Switzerland; [3]Klinik für Innere Medizin I, University Hospital Freiburg, Freiburg, Germany

**Abstract** In adult neurogenesis young neurons connect to the existing network via formation of thousands of new synapses. At early developmental stages, glutamatergic synapses are sparse, immature and functionally 'silent', expressing mainly NMDA receptors. Here we show in 2- to 3-week-old young neurons of adult mice, that brief-burst activity in glutamatergic fibers is sufficient to induce postsynaptic AP firing in the absence of AMPA receptors. The enhanced excitability of the young neurons lead to efficient temporal summation of small NMDA currents, dynamic unblocking of silent synapses and NMDA-receptor-dependent AP firing. Therefore, early synaptic inputs are powerfully converted into reliable spiking output. Furthermore, due to high synaptic gain, small dendritic trees and sparse connectivity, neighboring young neurons are activated by different distinct subsets of afferent fibers with minimal overlap. Taken together, synaptic recruitment of young neurons generates sparse and orthogonal AP firing, which may support sparse coding during hippocampal information processing.

DOI: https://doi.org/10.7554/eLife.23612.001

*For correspondence:
josef.bischofberger@unibas.ch

**Competing interests:** The authors declare that no competing interests exist.

## Introduction

In the adult hippocampus new neurons are continuously generated throughout life. Appropriate control of new synapse formation is not only critically important for the survival of the young cells but also for proper circuit function. During postnatal development, new synapse formation is initiated via activation of extrasynaptic NMDA receptors on dendritic shafts and the first glutamatergic synapses are believed to form as 'silent synapses' lacking postsynaptic AMPA receptors (*Durand et al., 1996*; *Engert and Bonhoeffer, 1999*; *Maletic-Savatic et al., 1999*). Similarly, in adult born neurons spine formation is dependent on NMDA receptors (*Mu et al., 2015*; *Sultan et al., 2015*). The first glutamatergic synapses are mostly silent synapses expressing NMDA receptors but no AMPA-receptors (*Chancey et al., 2013*). These early NMDA-only synapses are activated during learning and shape dendrite development as early as 1–2 weeks post mitosis (*Tronel et al., 2010*). Consistent with this notion, the time window between 1–3 weeks after cell division is not only important for spine formation, it also constitutes a critical period for NMDA-dependent survival, as cell death during this period is strongly increased in newborn neurons with a genetic deletion of the NR1 NMDA receptor subunit (*Tashiro et al., 2006*, *2007*; *Mu et al., 2015*). However, the functional impact of silent synapses in adult-born granule cells is largely unclear.

The AMPA component substantially increases during the following weeks and glutamatergic synapses induce AMPA-receptor dependent firing of APs after about 4 weeks post mitosis (*Mongiat et al., 2009*; *Dieni et al., 2013*). Glutamatergic AP firing in young granule cells might be important to generate synaptic output, but also to support activity-dependent synaptic plasticity at

input and output synapses (*Schmidt-Hieber et al., 2004*; *Ge et al., 2007*; *Bischofberger, 2007*; *Gu et al., 2012*). At this stage, however, firing of young cells has been proposed to be relatively unspecific. Due to reduced GABAergic inhibition and cellular 'hyperexcitability' the young population might be active under many different behavioural conditions, not discriminating between different memory items (*Marín-Burgin et al., 2012*; *Danielson et al., 2016*). On the other hand, it was claimed that synapse-evoked firing is difficult to obtain in newborn neurons up to 4 weeks post mitosis, due to low excitatory synaptic connectivity (*Dieni et al., 2016*).

Taken together, the current model suggests that glutamatergic synapses in adult-born hippocampal granule cells do not induce specific AP firing before being fully mature at about 6 weeks after mitosis. In contrast, behavioural data suggest the opposite, indicating a contribution of young neurons to hippocampus-dependent learning tasks already around ~3–4 weeks post mitosis (*Kee et al., 2007*; *Clelland et al., 2009*; *Kheirbek et al., 2012*; *Nakashiba et al., 2012*; *Gu et al., 2012*), probably by improving pattern separation. The reasons for this discrepancy between the apparent lack of specific AP firing and the contribution of new neurons to learning at this young age are unknown.

Here, we show that already at ~2–3 weeks newborn young granule cells generate input-specific sparse and orthogonal AP firing. At this stage they form 'silent' synapses which are dynamically unblocked during brief burst-activity in presynaptic fibers. This reliably induced NMDA-receptor dependent AP firing in the young neurons. Finally, we used paired patch-clamp recordings in neighbouring granule cells to show that the young neurons can be excited by a small non-overlapping population of afferent glutamatergic fibers, well suited to support sparse coding of neuronal information already from 2 weeks post mitosis onwards.

## Results

### Large NMDA-receptor dependent EPSPs in newly generated 2 week old granule cells

Newly generated young granule cells in the adult hippocampus form glutamatergic synaptic connections at about 2 weeks, mainly as functionally silent synapses (*Ge et al., 2006*; *Mongiat et al., 2009*; *Chancey et al., 2013*). To examine glutamatergic synaptic signaling in these neurons, we identified 2-week-old neurons in adult mice using retrovirus-mediated GFP labeling (*Figure 1*, *Figure 1—figure supplement 1*, *Zhao et al., 2006*). Previously, we have characterized the development of passive membrane properties in newborn granule cells showing that the input resistance strongly changes during the first 4–6 weeks after mitosis, exponentially decaying towards mature values of around 200 MΩ (*Heigele et al., 2016*). Similar to previous data, the $R_{in}$ of retrovirus labelled neurons ($n = 77$) was initially very high (~32 GΩ) at 7 days post injection (dpi), followed by a ~4 fold decay per week (*Figure 1—figure supplement 1*). As a second model system, we used transgenic mice expressing the red fluorescent protein DsRed under the control of the doublecortin (DCX) promoter, labeling young neurons within about 4 weeks post mitosis (*Brown et al., 2003*; *Couillard-Despres et al., 2006*; *Heigele et al., 2016*). The age of DCX-DsRed positive neurons was either classified as 2–3 weeks post mitosis ($n = 77$, range 2–8 GΩ) or 3–4 weeks post mitosis ($n = 27$, range 0.5–2 GΩ), based on the fitted exponential decay of $R_{in}$ in virus-labelled neurons (*Figure 1—figure supplement 1c–d*, see Materials and methods).

To examine glutamatergic synaptic transmission we stimulated presynaptic fibers in the molecular layer and sequentially recorded excitatory postsynaptic currents (EPSCs) and potentials (EPSPs) in young neurons in the presence of 100 µM picrotoxin to block GABAergic currents (*Figure 1*). For comparison mature neurons were recorded at the outer border of the granule cell layer using the same stimulation intensity (*Figure 1a,b*, ~30 µA). Whereas mature granule cells showed large AMPA-receptor mediated EPSCs at −80 mV ($1291 \pm 304$ pA, $n = 7$), the synaptic currents in 2 wpi neurons were more than 100-fold smaller ($3.3 \pm 1.0$ pA, $n = 6$). By contrast the difference in EPSP amplitude between mature and 2-week-old cells was much less pronounced ($13.9 \pm 2.8$ mV, $n = 14$ versus $3.2 \pm 0.4$ mV, $n = 6$) and the ratio between EPSP and EPSC was about 25-times larger at 2 wpi ($2.17 \pm 0.47$ mV/pA, $n = 6$) compared to mature neurons ($0.08 \pm 0.01$ mV/pA, $n = 19$, *Figure 1c*). Similarly, synaptic responses were obtained in DCX-DsRed expressing neurons with an estimated age of 2–3 weeks ($n = 12$), while neurons between 3–4 weeks post mitosis ($R_{in} = 0.5–2$ GΩ, *Figure 1c*, yellow, $n = 19$) showed intermediate AMPA-receptor mediated EPSC amplitudes

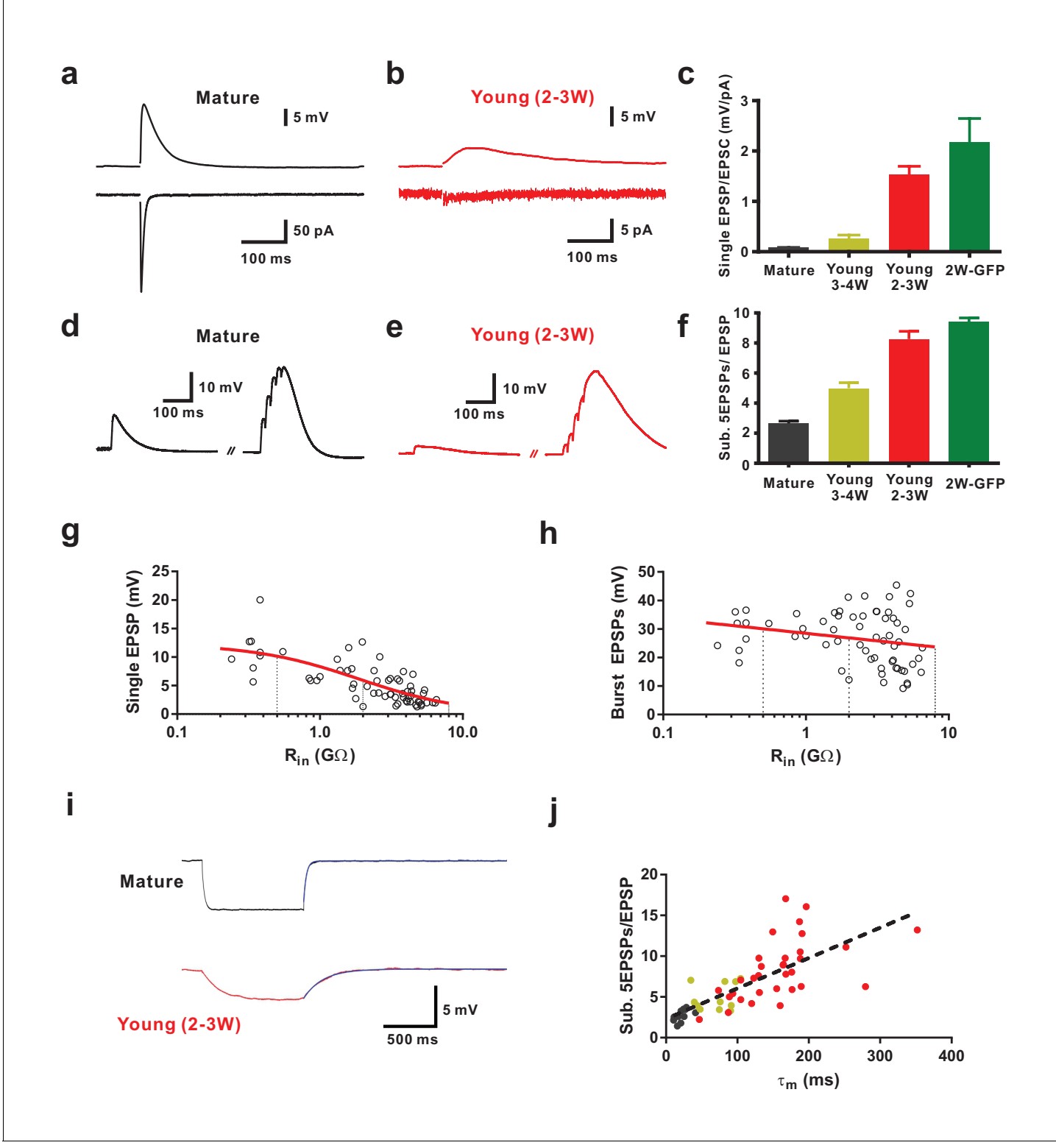

**Figure 1.** Efficient subthreshold EPSP summation in newborn young granule cells. (**a, b**) Example of synaptic EPSPs (top) and corresponding EPSCs (bottom) in a mature (**a**) and in a DCX-DsRed expressing young granule cell with an estimated age of 2.5 weeks post mitosis (**b**), recorded at resting membrane potential ($V_m$ = −80 mV). Extracellular stimulation intensity was 30 µA with 0.2 ms duration. (**c**) The ratio of synaptic EPSP relative to EPSC amplitude in individual cells is significantly higher in 2 wpi GFP labelled neurons (2W-GFP, p<0.001) and 2–3 week old DCX-DsRed labelled neurons (young 2–3W) relative to mature granule cells (p<0.0001, Mann-Whitney). The EPSP/EPSC ratio in 2W-GFP and young 2–3 W cells was not significantly
*Figure 1 continued on next page*

*Figure 1 continued*

different (p=0.282, Mann-Whitney). Bars for mature, 3–4W-DsRed, 2–3W-DsRed and 2W-GFP cells represent data from n = 19, 12, 8 and 6 neurons, respectively. (d, e) Subthreshold summation of five EPSPs evoked by brief burst stimuli (5@50 Hz, 20 µA) in a mature (d) and a young 2.5 week old GC (e). (f) The ratio of burst EPSP amplitude to single EPSPs is significantly higher in 2W-GFP (p<0.0001, Mann-Whitney) and young 2–3 week old neurons (p<0.0001, Mann-Whitney) relative to mature cells. Bars for mature, 3–4W young, 2–3W young and 2W-GFP cells represent data from n = 20, 33, 51 and 11 neurons, respectively. Stimulation intensity: 10–20 µA. (g, h) Amplitude of single EPSPs (g) and burst EPSPs (h, 5@50 Hz, 20 µA) in young and mature GCs were plotted against $R_{in}$ (n = 63). Single EPSPs (g) were fitted with a sigmoidal function, showing a half-maximal amplitude with $R_{in}$ = 1.9 GΩ. Burst EPSPs (h) were fitted with linear regression analysis revealing a small but non-significant decrease with $R_{in}$ (p=0.08). The vertical dashed lines indicate the $R_{in}$ at 4, 3 and 2 weeks post mitosis. (i) Examples of membrane potential hyperpolarization (~5 mV) by small negative current pulses in a mature (top) and young 2.5 week old GC (bottom). The blue curve represents a mono-exponential fit to the repolarization phase for estimation of the membrane time constant ($\tau_m$). (j) The ratio of burst EPSP amplitude to single EPSPs significantly correlates with the membrane time constant of young and mature GCs. Dashed line represents a linear regression, showing that the slope is significantly non-zero (p<0.0001, n = 55). The dots in black, yellow and red are data points from mature, 3–4 week old DsRed and 2–3 week old DsRed cells, respectively.

DOI: https://doi.org/10.7554/eLife.23612.002

The following figure supplement is available for figure 1:

**Figure supplement 1.** Identification of newborn granule cells in the adult mouse hippocampus.

DOI: https://doi.org/10.7554/eLife.23612.003

and EPSP-EPSC ratios. Taken together, the data indicate that glutamatergic synapses can effectively depolarize newly generated granule cells younger than 3 weeks, although AMPA-receptor mediated synaptic currents are tiny.

Synaptic activity in the dentate gyrus in vivo is rhythmically patterned at theta-gamma frequencies (*Pernía-Andrade and Jonas, 2014*). To study subthreshold postsynaptic integration during patterned synaptic activity, we applied brief bursts of 5 stimuli with low-intensity (20 µA) at approximately gamma frequency (50 Hz, *Figure 1d,e*). On average, newly generated cells at 2 wpi showed a burst EPSP amplitude of 30.2 ± 3.7 mV (n = 8) not significantly different from mature granule cells (29.8 ± 1.6 mV; n = 13, p=0.91, Mann-Whitney). Plotting the amplitude of single EPSPs in young and mature granule cells versus $R_{in}$ revealed a strong dependence on maturity (*Figure 1g*). Fitting the data with a sigmoidal decay indicates that EPSP size reaches 50% of mature levels at 1.9 GΩ, corresponding to an estimated age of about 3 weeks post mitosis. By contrast, plotting the burst amplitude versus $R_{in}$, was not significantly dependent on $R_{in}$ as suggested by linear regression analysis (p=0.08, *Figure 1h*). This is remarkable, as it shows that strongly different amplitudes of synaptic currents in young and mature cells generates the same burst EPSP amplitude. This indicates that the increase in synapse number is largely balanced by a concomitant (homeostatic) decrease in excitability and decrease in the impact of synaptic currents onto membrane depolarization.

In some mature and young neurons the burst stimulation evoked action potentials (APs). Therefore, these cells were omitted from the analysis above. To examine subthreshold integration in all granule cells, we adjusted stimulation intensity (~10–20 µA) to evoke burst EPSPs with a peak potential just below the AP threshold and quantified the ratio of burst EPSP versus single EPSPs in individual cells (*Figure 1f*). As single EPSPs were smaller in young cells, the ratio of the burst amplitude relative to single EPSPs was ~4 times larger at 2 wpi (9.57 ± 0.29, n = 11) compared to mature neurons (2.66 ± 0.14, n = 20, p<0.0001, *Figure 1f*). We further plotted burst–EPSP ratios against the membrane time constant measured in the very same neurons (*Figure 1i,j*), showing a significant correlation (p<0.0001, Spearman Correlation r = 0.83). This suggests that the high membrane resistivity and associated slow membrane time constant contribute significantly to the effective temporal summation of EPSPs during brief burst activity. These data show that an about 100-fold difference in EPSC-EPSP conversion in young neurons compensates for the much smaller synaptic currents to finally reach similar burst amplitudes in young and mature neurons.

Using voltage-clamp recordings we confirmed that new glutamatergic synapses onto newborn young granule cells express a relatively high density of NMDA receptors and show a large NMDA-AMPA receptor ratio (*Figure 2*, *Chancey et al., 2013*). Slow NMDA-receptor mediated synaptic currents in the absence of AMPA-receptor currents constitute the typical finger print of 'silent synapses'. This appears to be the predominant type of glutamatergic synapses in young neurons before 3 weeks of cell age (*Figure 2a–e*), whereas in mature neurons rapid AMPA-receptor currents dominate (*Figure 2a*). Furthermore, NMDA currents in young cells at 2 wpi are about ~50 times smaller

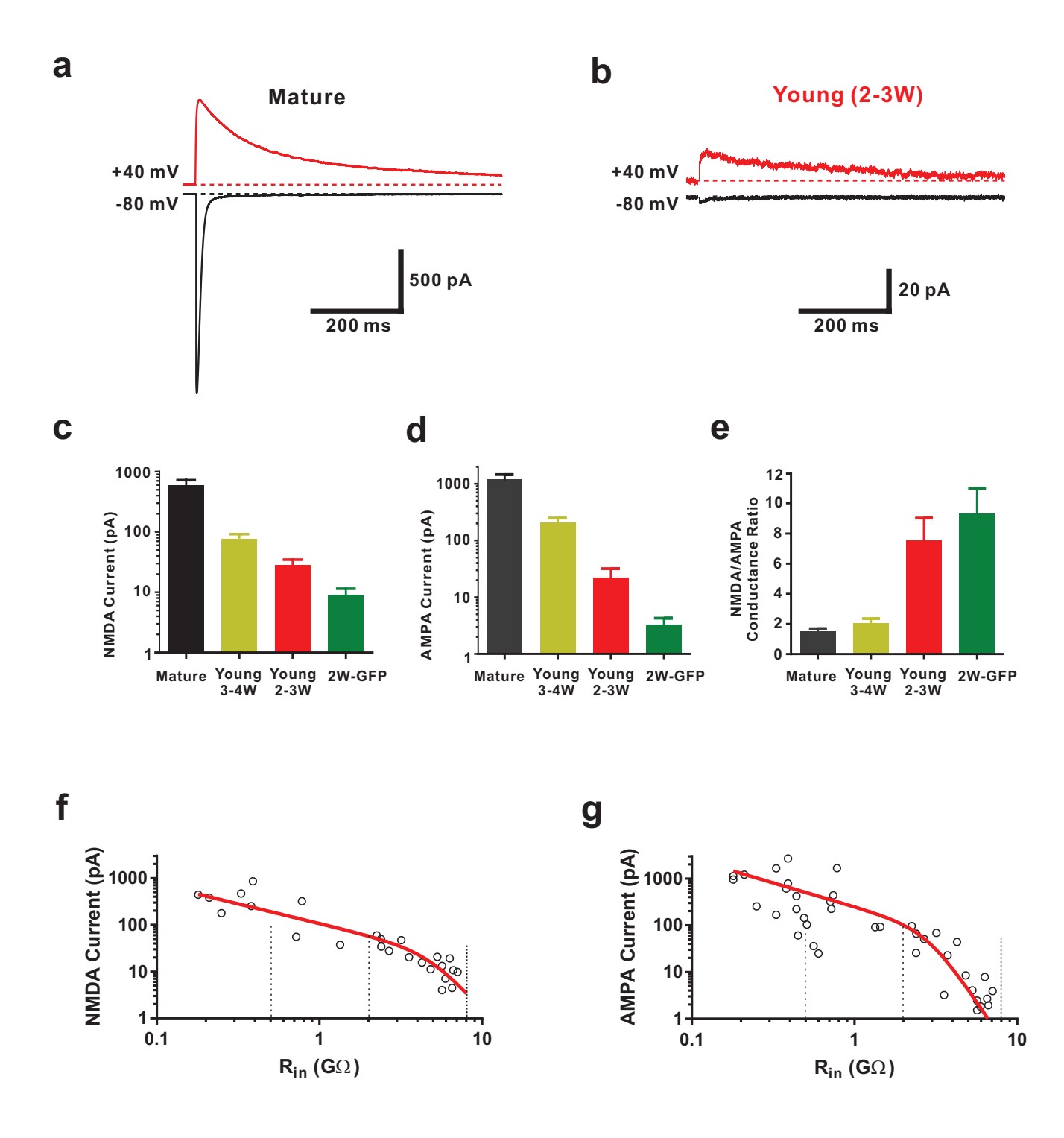

**Figure 2.** Functional properties of NMDA-receptor mediated EPSCs in young GCs. (**a, b**) Whole-cell voltage-clamp recordings of NMDA-receptor mediated EPSCs in presence of CNQX (top,+40 mV) and of CNQX-sensitive AMPA-mediated EPSCs (bottom, −80 mV) in a mature (**a**) and 2–3 week old DsRed neuron (**b**). (**c**) Bar graph showing the gradual increase of NMDA-receptor mediated EPSCs during the maturation of newborn GCs. All developmental stages are significantly different from each other (p<0.05, Mann-Whitney-Test). Bars for mature, 3–4W DsRed, 2–3W DsRed and 2W GFP cells represent data from $n$ = 7, 3, 12 and 6 neurons, respectively. (**d**) Bar graph showing the gradual increase of AMPA-receptor mediated EPSCs. All developmental stages are significantly different from each other (p<0.05, Mann-Whitney-Test). Bars for mature, 3–4W DsRed, 2–3W DsRed and 2W GFP

*Figure 2 continued on next page*

*Figure 2 continued*

cells represent data from *n* = 7, 3, 12 and 6 neurons, respectively. (**e**) Young 2W-GFP and 2–3 week old DsRed neurons show a significantly higher NMDA- to AMPA-receptor mediated conductance ratio relative to mature GCs (p<0.05). Young 2W-GFP and 2–3 week old DsRed neurons are not significantly different form each other (p=0.43, Mann-Whitney-Test). Bars for mature, 3–4W DsRed, 2–3W DsRed and 2W GFP cells represent data from *n* = 7, 3, 12 and 6 neurons, respectively. (**f, g**) Amplitude of NMDA currents recorded at +40 mV (f, *n* = 28) and AMPA currents recorded at −80 mV (g, *n* = 43) in young and mature GCs was plotted against log(Rin). The data were fitted with an exponential decay multiplied with a sigmoidal function (see Materials and methods). The vertical dashed lines indicate the $R_{in}$ at 4, 3 and 2 weeks post mitosis.

DOI: https://doi.org/10.7554/eLife.23612.004

(9.2 ± 2.3 pA, *n* = 6) compared to mature neurons (414 ± 83 pA, *n* = 7, p<0.01), consistent with a low synapse density at 2 weeks (*Figure 2c*, *Zhao et al., 2006*).

Given the relatively large NMDA-AMPA receptor ratio in young cells, we tested the contribution of NMDA receptors to small subthreshold EPSPs (*Figure 3*). Remarkably, blocking NMDA receptors reduced the EPSP amplitude in young neurons by 53.6 ± 6.3% (*n* = 9, p<0.0001), while there was only a small and non-significant effect in mature granule cells (15.5 ± 6.2%, *n* = 5, p=0.069, Mann-Whitney, *Figure 3a–c*). This was not due to a potential difference in voltage-dependent $Mg^{2+}$-block of NMDA-receptor mediated currents in young and mature neurons (*Figure 3d*). The relative open probability at −80 mV extracted from the current-voltage relationship was about ~3% of maximum in both, young and mature cells (*Figure 3de*, 3.2 ± 0.3% versus 3.3 ± 0.4% in 6 young and four mature cells, respectively, p=0.91, Mann-Whitney). Furthermore, NMDAR-gating kinetic was slightly different, showing an about 1.6-fold slower decay time course in young versus mature neurons, which might be explained by the expression of GluN2B subunits in young versus GluN2A subunits in mature granule cells (*Figure 3f*, *Chancey et al., 2013*). This indicates that, although there is a normal $Mg^{2+}$ block, the small but non-zero NMDA-conductance at resting membrane potential can depolarize young neurons due to high input resistance and slow membrane time constant.

These results show that the number of glutamatergic synapses at 2 weeks is low, forming immature NMDA-receptor expressing 'silent' synapses. However, the intrinsic properties of young neurons promote membrane depolarizations by very small NMDA currents at the resting membrane potential, resulting in dynamic unblocking of further synaptic NMDA-receptors during brief burst activity, finally leading to large synaptic potentials.

## Efficient NMDA-dependent spiking in young granule cells

In some young neurons, burst stimulation could evoke AP firing as already mentioned above. To systematically investigate the initiation of AP firing by NMDA-EPSPs, we varied extracellular stimulation intensity between 10–40 µA and performed current-clamp recordings in the presence of picrotoxin. The brief-burst stimulation (5@50 Hz) evoked APs in both, young and mature granule cells with 30 µA-stimulation intensity (*Figure 4*). Remarkably, the probability to evoke an AP via burst stimulation in cells with an estimated age of 2–3 weeks was only slightly smaller (88.4 ± 3.2%, *n* = 38, 30µA) compared to mature cells (100 ± 0%, *n* = 22, p=0.011, Mann-Whitney, *Figure 4f*), although the synaptic conductance was several orders of magnitude smaller. Similar results were obtained in 2-wpi virus labeled neurons (93.3 ± 3.8%, *n* = 12).

Although burst stimulation successfully evoked APs in both, young and mature neurons, several functional differences were obvious. First, young neurons showed typically a single postsynaptic spike, whereas mature cells fired on average about three spikes per burst (*Figure 4ab*). Second, spike amplitude was smaller in young neurons (108.5 ± 2.5 mV, *n* = 21) than in mature cells (143.3 ± 1.3 mV, *n* = 5, p<0.0001). Third, half-duration of spikes was longer in young (5.2 ± 0.8 ms, *n* = 21) versus mature cells (1.6 ± 0.1 ms, *n* = 5, p<0.0001). These results are consistent with the notion that the density of voltage-gated sodium and potassium channels is lower in the young versus mature neurons. Finally, spiking in young cells started with longer delay (98 ± 9 ms, *n* = 21) than in mature cells (45 ± 7 ms, n = 5, p<0.0001), relative to stimulation onset, consistent with the slow membrane time constant and prolonged temporal summation of EPSPs in young neurons. Notably, single stimuli with high intensity could evoke APs in mature granule cells, but never induced spiking in young neurons with $R_{in}$ above 2 GΩ (data not shown).

Whereas application of CNQX blocked burst-evoked AP firing in mature neurons, spiking was largely unaffected in 2- to 3 week old cells generating 1.28 ± 0.11 and 1.34 ± 0.15 APs per burst in

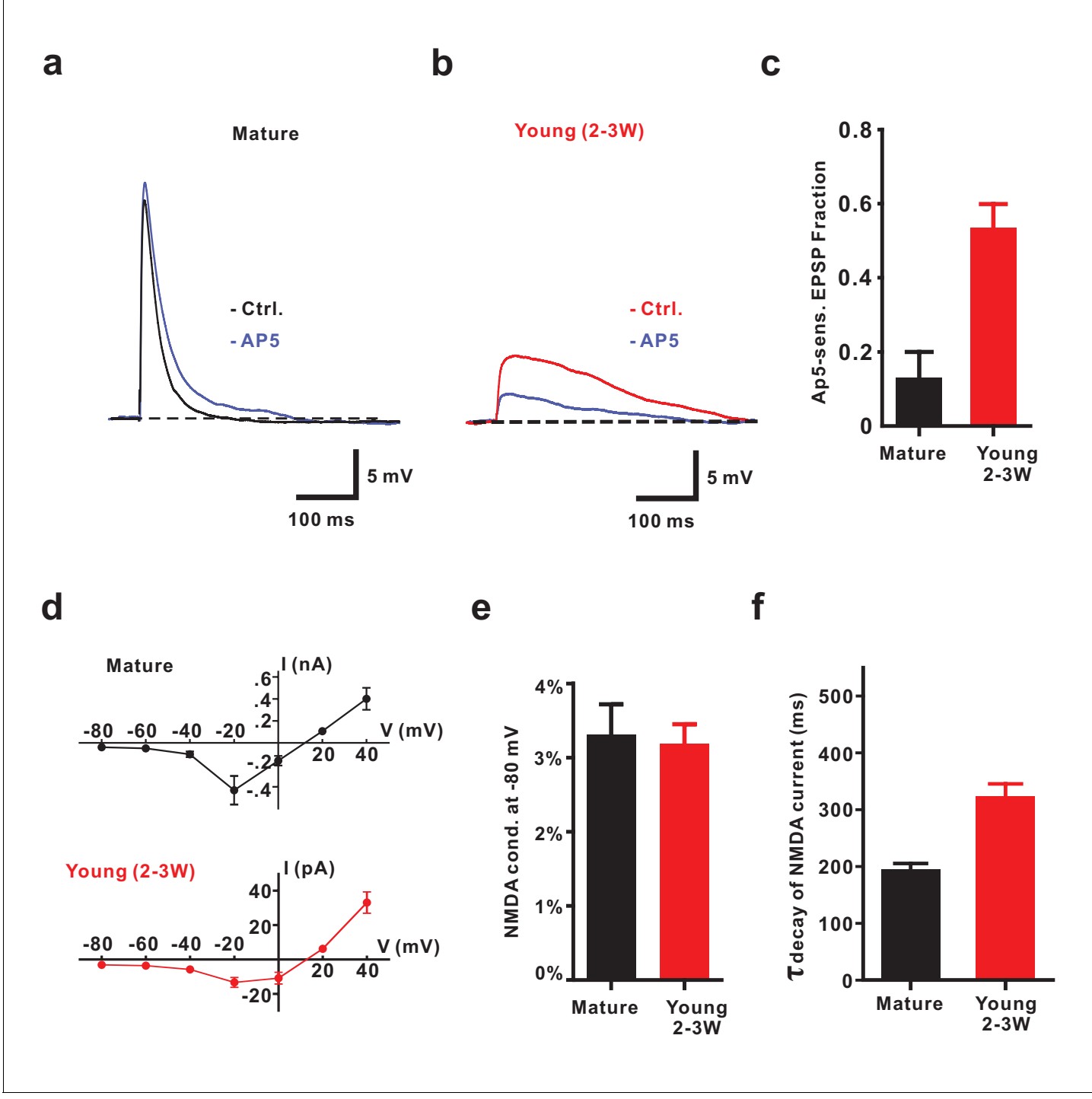

**Figure 3.** NMDA receptors contribute to EPSPs from resting membrane potential in young GCs. (a, b) Application of 50 μM AP5 does not significantly affect EPSP amplitude in mature GCs (a) but decreases EPSP amplitude in young neurons (2–3W DsRed, (b). (c) Bar graph showing a significantly larger contribution of AP5-sensitive NMDA receptors to the EPSP in 2–3 week old DsRed neurons than in mature GCs (p<0.01, Mann-Whitney-Test). In contrast to the mild effect of AP5 in mature GCs (p=0.31, Wilcoxon Matched Pairs Test, $n$ = 5), AP5 application markedly decreased the EPSP amplitude in young GCs (p<0.01, Wilcoxon Matched Pairs Test, $n$ = 9). (d) I-V relationship of synaptic NMDA-receptor mediated currents in mature ($n$ = 4, top) and young GCs ($n$ = 7, bottom). NMDA receptors in both groups elicited maximal inward currents at −20 mV. (e) The ratio of the NMDAR-mediated conductance at −80 mV relative to the conductance at +40 mV was not significantly different between mature ($n$ = 4) and young GCs ($n$ = 6, p=0.91, Mann-Whitney). (f) NMDA-receptor mediated EPSCs (recorded at +40 mV) decay significantly slower in young ($n$ = 11) than in mature GCs ($n$ = 7, p<0.001, Mann Whitney).

DOI: https://doi.org/10.7554/eLife.23612.005

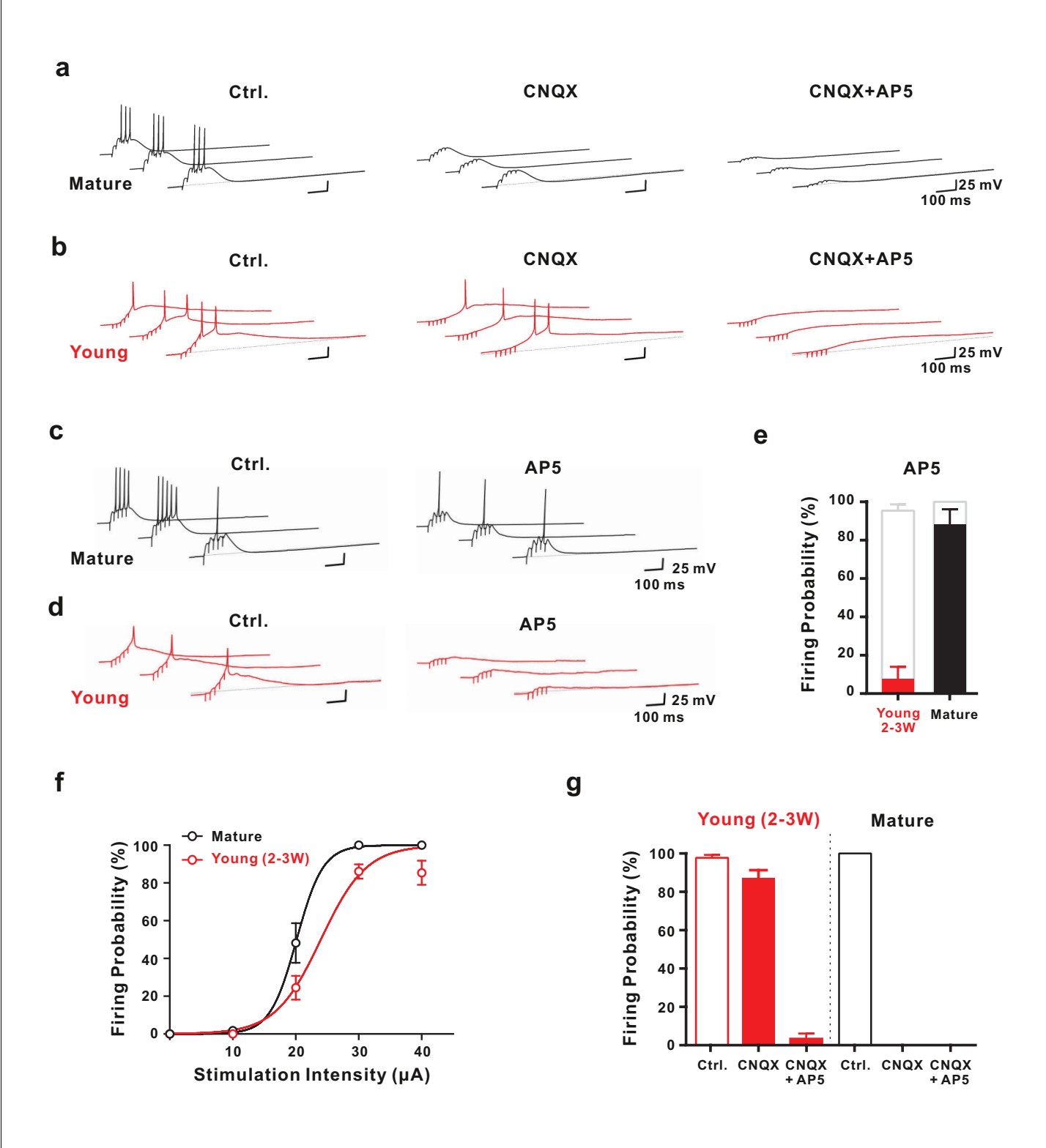

**Figure 4.** Synaptic NMDA-receptor dependent spiking in young GCs. (a,b) Synaptically evoked APs in mature GCs (a) were blocked by CNQX. By contrast in young GCs (b), firing was insensitive to CNQX, while it was fully blocked by AP5 (30 µA). (c,d) Conversely, in AP5 alone firing was still possible in mature neurons (c), but was largely blocked in young GCs (d). (e) The bar graph summarizes the strong reduction in AP success rate per burst in young cells (7.7 ± 6.2% versus 95.4 ± 3.3% in control, *n* = 13), as well as the small effect in mature GCs (88.0 ± 8.0% versus 100% in control, *n* = 5). (f) The probability to fire one or more spikes per burst was plotted against stimulation intensity, showing that 2–3 week old DsRed neurons
*Figure 4 continued on next page*

*Figure 4 continued*

(*n* = 38) reach AP threshold as efficiently as mature GCs (*n* = 22) during burst stimulation. The data were fitted with a sigmoidal function (black and red curves), with half-maximal firing rate at a stimulation intensity of 20.2 μA and 23.6 μA in mature and 2–3 week-old neurons, respectively. (g) Bar graphs showing firing probability of the young (red) and mature GCs (black) under pharmacological conditions described in (a) and (b). Unlike the dominant effect of CNQX on firing in mature GCs (control vs. CNQX: from 100% to 0, *n* = 5), it had only minor effects in young GCs (control vs. CNQX: from 98% to 87%, p<0.05, Wilcoxon Matched Pairs Test, *n* = 17). Additional AP5 application effectively eliminated the firing in young GCs. The red bars represent data from *n* = 5 two-week-old GFP and *n* = 12 DCX-DsRed 2–3 week old cells.

DOI: https://doi.org/10.7554/eLife.23612.006

control and CNQX, respectively (p=0.69, *n* = 16, *Figure 4b*). Similarly, spike half-duration and peak amplitude was not significantly affected. The only difference was a ~50% larger spike latency relative to burst onset in CNQX (172 ± 15 ms vs 98 ± 10 ms, *n* = 16). However, the most important component in young cells is provided by NMDA receptors, as application of AP5 largely blocked AP generation (*Figure 4c–e*). After application of both, CNQX and AP5, a residual small and slow subthreshold depolarization remained (*Figure 4b*). As we were not able to block this component, we could not test or rule out a potential contribution to voltage-dependent activation of NMDA receptors in the young neurons. By contrast, mature granule cells still fired APs in AP5 in response to burst stimulation with high success rate (*Figure 4e*). The number of spikes per burst, however, was reduced (1.12 ± 0.23 versus 3.16 ± 0.39, *n* = 5, p=0.0011).

These results show that AP firing on fully connected mature granule cells is critically dependent on strong AMPA-receptor activation with some modulation by NMDA receptors. By contrast, in young neurons NMDA receptors are critically important and silent synapses can be dynamically unsilenced by brief-burst activity to reliably evoke AP firing in absence of AMPA receptors. Again, this supports the conclusion, that the lower synapse number in young cells is balanced by the enhanced intrinsic excitability. Remarkably, this can generate glutamatergic AP firing in young cells with similar probability as in mature granule cells despite a more than 100-fold difference in EPSC amplitude at resting membrane potential.

## Selective activation of young granule cells by a low number of NMDA synapses

As the young cells are very sensitive to small glutamatergic currents, they might respond non-specifically to diffuse populations of active presynaptic fibers. On the other hand, the young neurons show a smaller dendritic tree which may restrict the number of potential presynaptic partners (*Zhao et al., 2006*). To address this topic we first quantified the morphology of the dendritic tree of biocytin-filled young and mature granule cells, which showed a cone-like shape (*Figure 5*). The diameter of the dendritic cone was on average 4-fold smaller in young (86 ± 15 μm, *n* = 8) versus mature neurons (324 ± 17 μm, *n* = 7, p=0.0003). Furthermore, the height of the cone was smaller in young (127 ± 13 μm) than in mature cells (188 ± 12 μm, p=0.0037) suggesting that the total cone volume is substantially smaller in the young neurons. This indicates that the chance that an entorhinal afferent fiber might target a specific young neuron is much smaller than for a given mature cell. To assess the probability by which glutamatergic afferent fibers might successfully activate adult–born young granule cells, we varied the location of the stimulation electrode tangentially in the molecular layer at a fixed radial distance at about 50–100 μm from the granule cell layer to activate different subsets of presynaptic fibers (*Figure 6*). As shown in *Figure 6a–c*, stimulation sites that successfully induced spike discharge were spatially much more restricted in DCX-DsRed positive young neurons relative to mature granule cells. Although young neurons could fire very successfully in response to proximal stimulation, the firing probability quickly fell below 50% after tangentially moving the stimulation electrode by a few tens of micrometers. On average, the half-width of the dendritic 'firing field' was about 5-times more narrow in young neurons than in mature granule cells (76.7 ± 8.1 μm versus 360.8 ± 70.6 μm in *n* = 15 and *n* = 11 cells, p<0.0001, *Figure 6d*). The broad firing field in mature cells is consistent with a large amplitude of synaptic currents, a relatively large dendritic tree and several thousand spines and excitatory glutamatergic synapses present on dendrites of a mature granule cell (*Zhao et al., 2006*; *Schmidt-Hieber et al., 2007*). By contrast, the efficient conversion of small EPSCs into membrane depolarization in young cells allows for glutamatergic AP firing induced by the small subset of presynaptic fibers synapsing onto the small dendritic tree of these neurons.

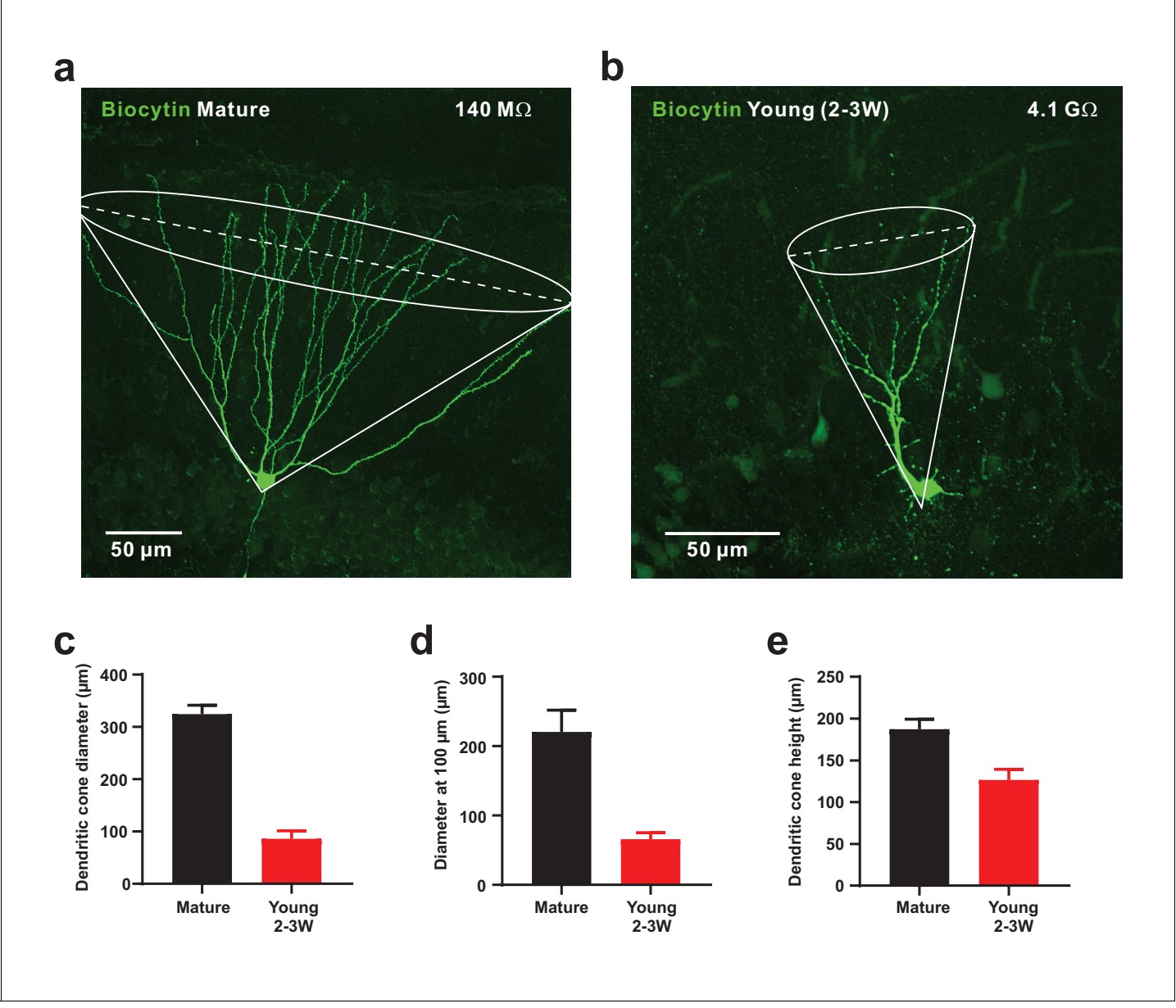

**Figure 5.** Small diameter of dendritic cones in young GCs. (**a, b**) Morphology of biocytin-filled mature (**a**) and young (**b**) granule cells. The somata are located at the outer and inner border of the granule cell layer, respectively. The cone overlay indicates the geometrical parameters measured, including cone diameter shown as dashed line. (**c–e**). Small diameter of dendritic cone (**c**) in young GCs (n = 8) compared to mature neurons measured at the base. Similarly, the cone diameter was measured at a distance of 100 µm from the soma (**d**). Additionally, the height of the cone is smaller in young versus mature GCs (**e**).

DOI: https://doi.org/10.7554/eLife.23612.007

The narrow 'firing field' might suggest that young neurons receive synaptic inputs from a common small subset of afferent fibers, preferentially targeting this young population. On the other hand, individual young neurons might be activated by different small subsets of presynaptic inputs. To distinguish between these possibilities, we used paired whole-cell patch-clamp recordings to analyze spatial sampling of the afferent neuronal input space by neighboring mature and neighboring young neurons (*Figure 7*). We have selected either two DsRed labeled young neurons close to the hilus, or two mature neurons at the outer border of the granule cell layer with a similar tangential soma distance of 52 ± 2 µm (*n* = 5 young pairs) and 54 ± 11 µm (*n* = 4 mature pairs, p=0.90, Mann-

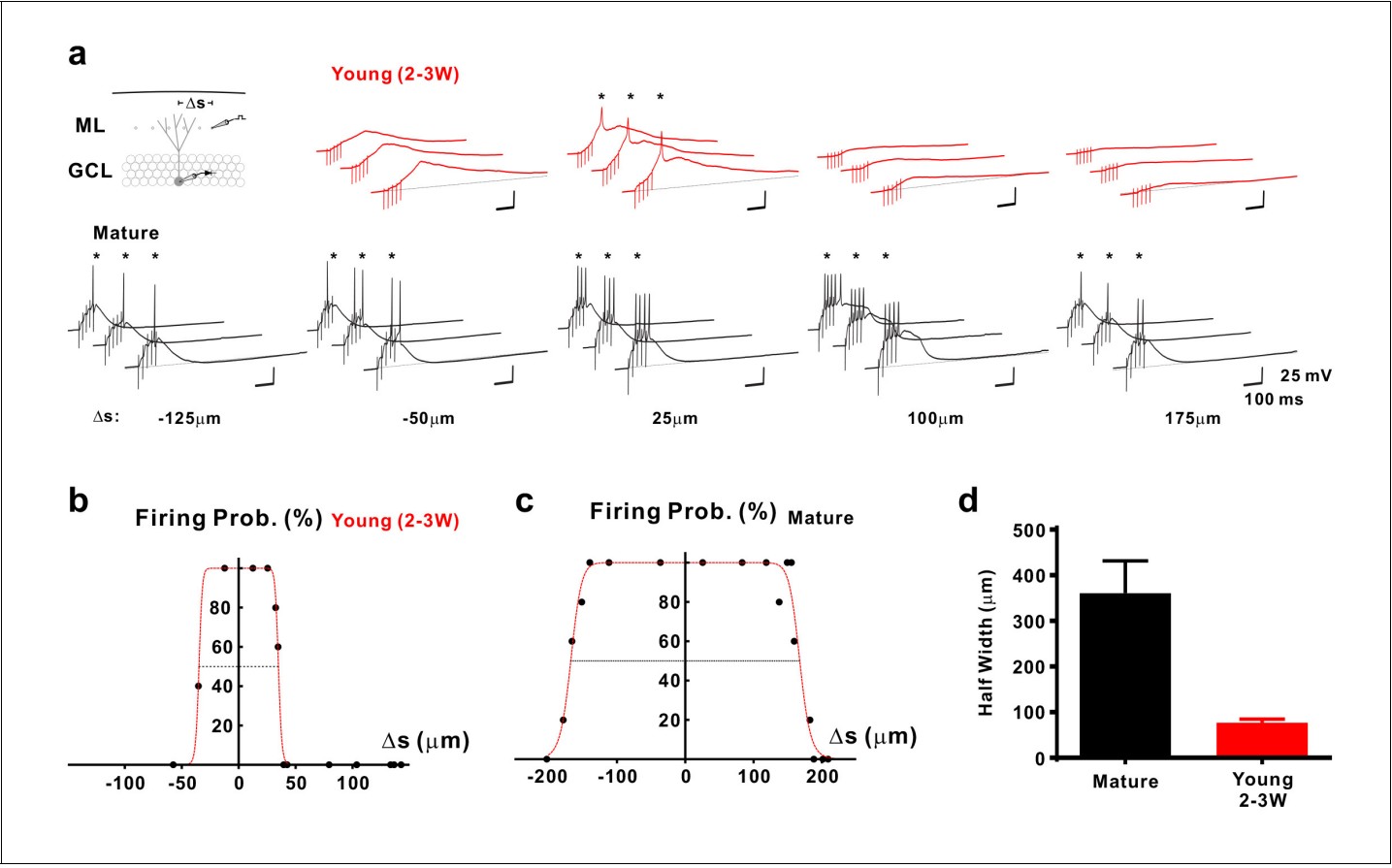

**Figure 6.** Selective sampling of glutamatergic presynaptic input space in young GCs. (**a**) Synaptically evoked spiking in a 2–3 week old neuron (top) and a mature GC (bottom) with extracellular stimulation at different locations. The stimulation electrode was tangentially shifted in the middle part of the molecular layer (ML) with constant distance of ~100 μm from the granule cell layer (GCL). The schematic diagram of the experimental configuration (top) shows the recorded cell (grey) and the stimulation positions (dots in the ML). Δs represents the distance from the stimulation position to the centre of the spatial range where stimulation generates 100% firing probability. APs are highlighted with stars (*). (**b,c**) The firing probability of the young (**b**) and mature cell (**c**) in (**a**) top and bottom, respectively, was plotted against location Δs of the stimulation electrode and fitted with a bell-shaped function (see Materials and methods for details). (**d**) Bar graph showing the mean diameter (half-width) of the dendritic 'firing fields' where APs are generated with more than 50% probability in mature (n = 11) and young (n = 15) GCs. The half-width in young GCs is significantly more narrow than in mature GCs (p<0.0001, Mann Whitney).

DOI: https://doi.org/10.7554/eLife.23612.008

Whitney), respectively. Whereas in neighboring mature granule cells the synaptic firing fields are largely overlapping (82.9 ± 7.2%, n = 8, *Figure 7a,c*), the overlap was much smaller in the young neurons (16.1 ± 7.3%, n = 10, p<0.001, *Figure 7b–d*), suggesting sparse synaptic connectivity of the young cells. This shows that young neurons sample activity in a spatially more restricted population of presynaptic fibers than mature cells, which leads to a distinct activation profile of neighboring young cells.

We have previously shown that depolarizing GABAergic synaptic inputs in young neurons can either facilitate or inhibit the generation of APs (*Heigele et al., 2016*). Thus, GABAergic synapses might change firing output. Therefore, we performed similar experiments in the absence of picrotoxin with GABAergic transmission intact (*Figure 8*, ACSF). While recording from a young neuron, glutamatergic fibers were either stimulated in the center of the dendritic firing field or with a second stimulation electrode laterally shifted by 100 μm (5@50 Hz). In ACSF, firing in young cells was only observed with center stimulation (*Figure 8a*, left), similar to the stimulation in gabazine, which was applied afterwards to the same cell (lower traces). At the 100 μm stimulation site we measured always some depolarization, but firing was largely absent, consistent with the absence of dendrites and glutamatergic inputs. Feedforward-activation of GABAergic interneurons at 100 μm might

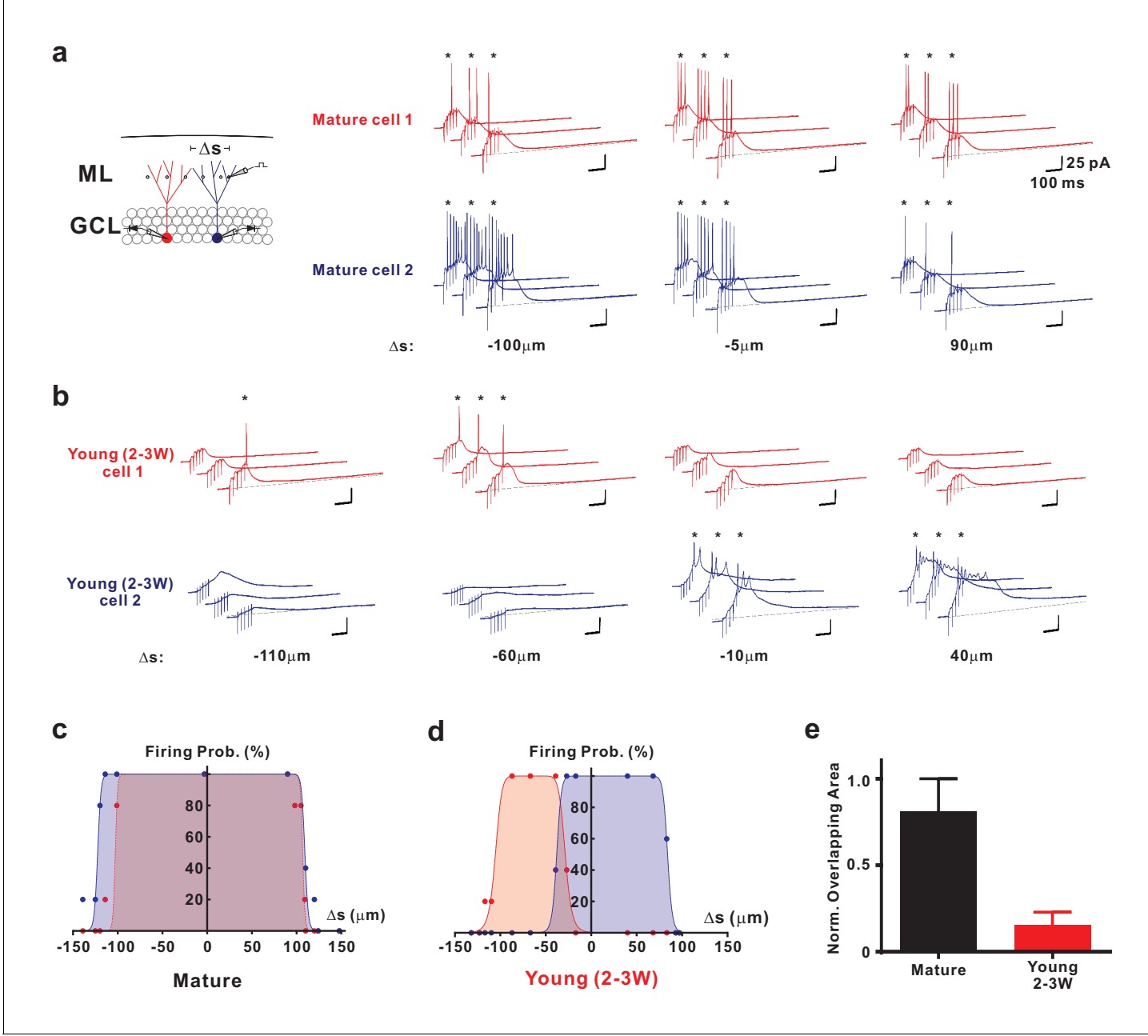

**Figure 7.** Distinct activation of different young neurons by different afferent fibre subpopulations. (**a**) Synaptically evoked spiking in two simultaneously recorded mature GCs. The stimulation electrode was positioned at different locations Δs along a track parallel to the granule cell layer as indicated. The schematic diagram of the experimental configuration (left) shows the two recorded cells (red and blue) and the stimulation positions (dots in the ML). Δs represents the distance from the stimulation position to the mid-line between the two cells. (**b**) Synaptically evoked spiking in two simultaneously recorded young GCs 2–3 weeks post mitosis at various stimulation positions in the ML. Please note, that the locations to evoke successful firing in two young neurons do not overlap. (**c, d**) The firing probability of the two simultaneously recorded mature (**c**) and young neurons (**d**) shown in a and b, respectively, was plotted against the location Δs of the stimulation electrode. The data were fitted with a bell-shaped function (see Materials and methods for details). Note the small overlap of the area under the curves generated from the young cells (**d**). (**e**) Bar graph showing the normalized overlap of the area under the curve in pairs of mature (*n* = 8) and young neurons (*n* = 10, p<0.001, Mann Whitney).

DOI: https://doi.org/10.7554/eLife.23612.009

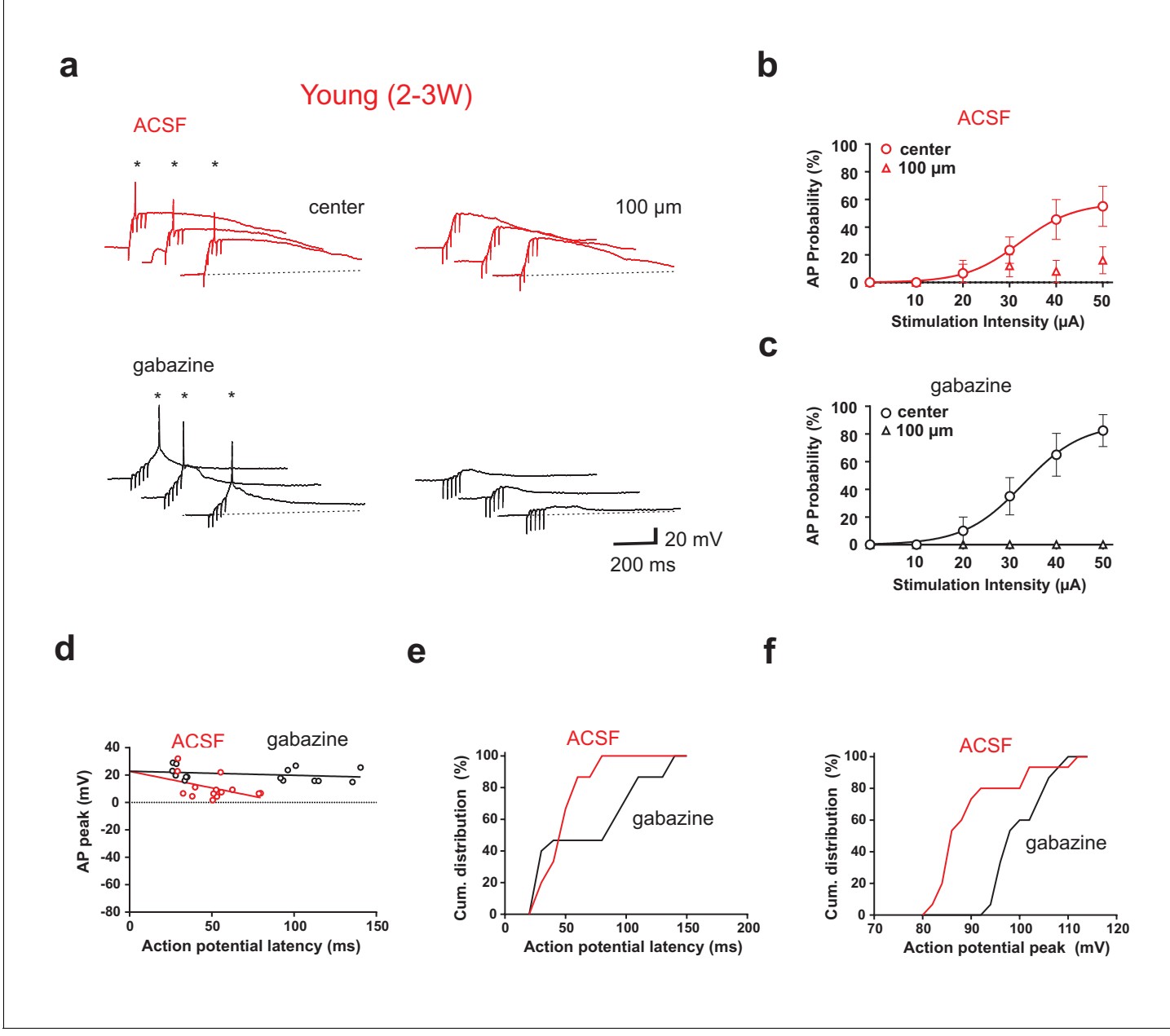

**Figure 8.** GABAergic synaptic transmission improves temporal precision of young GC firing. (**a**) In ACSF, brief burst stimulation (5@50 Hz) evoked APs in a young GC with the stimulation electrode located in the ML at the center of the dendritic firing field (upper left) but not when the electrode was tangentially shifted by 100 µm (upper right). Similar results were obtained in the presence of gabazine to block GABAergic transmission (lower traces, 50 µA). (**b**) Action potentials were evoked at different stimulation intensities in $n = 6$ out of 9 young cells with center stimulation (circles) but not with a tangentially shifted electrode (triangles). (**c**) Similarly, in the presence of gabazine AP firing in young neurons was restricted to center stimulation ($n = 9$). (**d**) The peak amplitude of the APs from experiments in (**b, c**) was plotted versus AP latency relative to stimulus onset and fitted by linear regression, indicating that AP latencies are shorter in ACSF (50 µA). The slope of the regression line was −24 ± 13 mV/100 ms in ACSF versus −3 ± 3 mV/100 ms in gabazine. (**e, f**) Cumulative distribution of AP latency (**e**) and peak amplitude (**f**) showing that the latency is significantly shorter (p<0.028, Kolmogorov-Smirnov) and the peak amplitude is significantly smaller (p<0.0001, Kolmogorov-Smirnov) in ACSF.

DOI: https://doi.org/10.7554/eLife.23612.010

The following figure supplement is available for figure 8:

**Figure supplement 1.** GABAergic interneurons strongly decrease burst-evoked AP firing in mature GCs.

DOI: https://doi.org/10.7554/eLife.23612.011

generate a dominant shunting effect via depolarizing lateral shunting inhibition (*Heigele et al., 2016*). This spatial profile was similar with different stimulation intensities ranging from 10 to 50 μA (*Figure 8bc*, *n* = 6 of 9 young cells). Furthermore, spike probabilities were similar in ACSF and gabazine.

Nevertheless, several parameters were affected by GABAergic synaptic transmission. First, spike latency with center stimulation was shorter in ACSF than in gabazine (48.8 ± 7.8 versus 77.1 ± 20.0 ms, *n* = 6, *Figure 8d*) with a specific absence of late spikes as shown by the cumulative distribution of latencies (*Figure 8e*, p=0.028, Kolmogorov-Smirnov). The time window for spikes was more restricted in ACSF (range 29–79 ms) while the range was much larger in gabazine (26–140 ms), well exceeding beyond the end of the burst stimulation (80 ms). Second, the peak amplitude of APs was on average smaller in ACSF than in gabazine (91 ± 2 mV versus 101 ± 1 mV, *n* = 6, p<0.0001 Kolmogorov-Smirnov, *Figure 8f*). Finally, the AP peak amplitude decreased with increasing latency (*Figure 8d*). These data indicate that initially (<50 ms) GABAergic depolarization might facilitate NMDA receptor activation. However, due to further increase in GABAergic synaptic conductance during the burst, the larger GABAergic conductances lead to reduced spike amplitude and finally (>80 ms) to shunting inhibition of spiking, similar to what was described previously (*Heigele et al., 2016*). Therefore, the slow time course of GABAergic synaptic transmission in adult-born young granule cells creates a window for early excitation followed by delayed shunting inhibition. In *n* = 3 out of 9 cells no spikes were evoked in ACSF, which might be related to the fact, that the center position was more difficult to find in ACSF than in picrotoxin.

Similar experiments were performed in mature granule cells (*Figure 8—figure supplement 1*) showing that intact inhibition strongly reduces firing in ACSF in *n* = 6 mature cells (*Figure 8—figure supplement 1a–c*). In 2 out of 6 mature cells no spikes could be evoked in ACSF. Moreover, spiking in mature cells is reduced in ACSF, independent of the localization of the stimulation electrode. In fact the difference between gabazine and ACSF appears to be larger (5.7 ± 0.5 versus 1.1 ± 0.2 APs per burst, center) than the difference between location of stimulation electrodes in ACSF (1.1 ± 0.2 versus 0.3 ± 0.2 APs per burst, *n* = 4). By contrast, the opposite is true for young neurons (*Figure 1—figure supplement 1ef*). These data are consistent with the well-known role of GABAergic inhibition in restricting perforant-path evoked AP firing in mature dentate granule cells (*Ewell and Jones, 2010*; *Dieni et al., 2013*; *Lee et al., 2016*).

In summary, young neurons show effective AP firing in response to brief burst activity in a low number of presynaptic glutamatergic fibers due to dynamic un-silencing of 'silent' synapses already at 2 weeks post mitosis. Furthermore, the firing of the young neurons at this age is restricted to small subpopulations of active presynaptic fibers, distinguishing them from the much broader excitatory synaptic connectivity in mature granule cells.

## Discussion

Here we show that glutamatergic synapses onto newly generated young granule cells are formed as 'silent' synapses containing mainly NMDA receptors. In strong contrast to the canonical model of synapse formation, we unexpectedly found that 'silent' glutamatergic synapses can initiate AP firing in newborn young granule cells at early developmental stages. We identified several key factors important for input-specific AP firing with silent synapses. First, although there is a normal $Mg^{2+}$-block of NMDA receptors, these receptors can generate EPSPs from resting membrane potential due to the high electrical input resistance in young cells. Second, brief bursts of a few presynaptic spikes generate large temporal summation of PSPs, sufficient to unblock further NMDA receptors generating a large dynamic increase in synaptic gain to induce AP firing. As the neurons mature, the increase in synapse number is balanced by a gradual decrease in excitability and impact on membrane depolarization. Third, GABAergic synaptic transmission shortens the time window for burst-evoked firing via depolarizing shunting inhibition. Finally, systematically activating different sets of afferent fibers indicates that firing of young cells is not unspecific, but rather dependent on small non-overlapping populations of afferent input fibers, due the sparse connectivity combined with high synaptic gain, generating differential and orthogonal spike output in neighboring young cells.

## AP firing by silent synapses

The first glutamatergic synapses onto newly generated young granule cells are formed around 2 weeks post mitosis (*Ge et al., 2006*; *Zhao et al., 2006*; *Chancey et al., 2013*). As in most developing young neurons, these first synapses show functional characteristics of typical silent synapses (*Chancey et al., 2013*). Morphologically, young granule cells rapidly grow dendrites throughout the molecular layer during the first 3 weeks post mitosis (*Zhao et al., 2006*; *Gonçalves et al., 2016*). They show thin filopodia-like protrusions, which are believed to form immature contacts with pre-existing afferent glutamatergic fibers (*Toni et al., 2007*). Based on these data, glutamatergic (silent) synapses at 2–3 weeks were deemed to be insufficient for spike initiation (*Mongiat et al., 2009*; *Dieni et al., 2013*; *Chancey et al., 2013*). By using brief burst stimulation (5@50 Hz), we found that glutamatergic synapses can reliably induce action potentials even in 2- to 3-week-old neurons similar to mature neurons, given that the appropriate afferent fibers are stimulated. This is surprising, as not only the AMPA conductance at these synapses is tiny and negligible, but also the NMDA-receptor mediated currents are much smaller than in mature neurons (~50 times lower at 2 wpi).

The canonical model of NMDA-receptor activation at cortical glutamatergic synapses is based on a sequential activation of AMPA receptors, depolarizing the membrane to relieve voltage-dependent $Mg^{2+}$ block of NMDA receptors. The adult-born young granule cells show two major differences to that model. First, although there is normal $Mg^{2+}$ block, the NMDA receptors contribute to EPSPs at resting membrane potential (around $-80$ mV) because the small 3% $Mg^{2+}$-resistant fraction of the NMDAR-mediated conductance is sufficient to depolarize young cells due to the high input resistance (2–8 GΩ). Second, temporal EPSP summation during burst activity dynamically unblocks the remaining NMDA receptors. This is a consequence of the slow membrane time constant of 2- to 3-week-old granule cells and means that AMPA receptors are not necessary for burst-evoked AP firing if the postsynaptic input resistance is high enough. Therefore, the synapses are largely 'silent' with isolated single stimuli but 'speak up' with brief repetitive afferent activity of a few presynaptic APs at around gamma frequency (~50 Hz). The requirement for brief burst activity does not allow for spike-to-spike transmission. By contrast, this will filter out low frequency input signals and will restrict spike output to young cells getting highly active afferent glutamatergic synaptic inputs.

Silent synapses are the major type of glutamatergic input at early embryonic and postnatal developmental stages (*Liao et al., 1995*; *Isaac et al., 1995*; *Durand et al., 1996*). Our results show that synaptic silence is not as stereotypic as previously thought and appears to be highly context dependent. Given that the input resistance of young neurons can be relatively high, AP firing by silent synapses might also apply to different types of neurons during early brain development.

## Connectivity with afferent fibers

Interestingly, the burst EPSP amplitude as well as the firing probability in young neurons was similar to mature cells using the same stimulation intensity. This would indicate that a similar population of afferent glutamatergic fibers were activated and responsible for spike initiation in both cell types. However, the connectivity profile between afferent fibers and postsynaptic neurons is very different in young and mature granule cells. The amplitude of synaptic NMDA-receptor mediated currents at 2 wpi was about 2% relative to mature cells, increasing up to ~15% at 3 weeks. Assuming that the density of synaptic NMDA-receptors is about the same in immature (silent) and mature hippocampal synapses (*Takumi et al., 1999*), this would indicate, that the density of synapses is much more sparse at 2–3 weeks post mitosis (range 2–15%) than in mature neurons. This is consistent with morphological data showing that the density of dendritic protrusions (including filopodia and spines) is about ~25% at 3 weeks post mitosis relative to the density of protrusions in mature neurons (*Zhao et al., 2006*; *Toni et al., 2007*).

The data are also in line with previous physiological studies showing that spiking in 4-week-old immature granule cells with high Rin is limited by a small amplitude of excitatory synaptic currents in response to perforant-path stimulation (*Dieni et al., 2013*). This effect was even more pronounced with stimulation in the entorhinal cortex, generating much smaller excitatory synaptic currents in newborn 4-week-old cells than in mature neurons. As a consequence spiking probability was lower, suggesting that low synaptic connectivity prevents immature neurons from responding broadly to cortical activity, potentially enabling excitable immature neurons to contribute to sparse and orthogonal neuronal representations (*Dieni et al., 2016*). Consistent with these ideas we have found that

successive lateral movement of the stimulation electrode in the molecular layer lead to successful stimulation in a spatially much more restricted area in young as compared to mature granule cells. The 5-times smaller 'firing fields' in young cells nicely fit with morphological analysis showing a smaller dendritic cone diameter (*Figure 5*), less dendrites and a 4-fold smaller total dendritic length at 2 wpi (~500 μm) as compared to mature granule cells (~2000 μm, *Zhao et al., 2006*; *Schmidt-Hieber et al., 2007*; *Gonçalves et al., 2016*).

Using paired recordings from two adjacent young or mature neurons with comparable distance, young granule cells showed very small overlap when different populations of afferent fibers were activated. The minimal overlap between 'firing fields' in two adjacent young cells indicates that individual cells probe for activity in different non-overlapping sub-populations of afferent fibers. The requirement for brief-burst activity in presynaptic fibers will further sparsify the neuronal output signal in the young granule cell population. This suggests that different activity patterns in afferent fibers will generate sparse and highly different (orthogonal) output patterns in the young neuron population. Furthermore, our data indicate that silent-synapse induced firing can be fine-tuned by newly formed GABAergic synapses at this early developmental stage (*Heigele et al., 2016*). The gradually increasing GABAergic conductances during burst inputs temporally constrains spiking output in young neurons by restricting the time window for firing via early excitation and late shunting inhibition - without much change in total number of spikes.

Taken together, this indicates that sparse and orthogonal firing in young neurons is much more stereotyped and robust, while firing in mature cells potentially could be sparse, but is much more dependent on the complex interplay of thousands of glutamatergic afferents with a large population of local GABAergic interneurons (*Ewell and Jones, 2010*; *Dieni et al., 2013*; *Lee et al., 2016*).

## Functional implications for synapse formation and information processing

What is the functional role of sparse and orthogonal firing of newborn granule cells at this early developmental stage? Behavioral studies suggest that adult neurogenesis favor the disambiguation of similar memory items by promoting pattern separation (*Sahay et al., 2011*; *Kheirbek et al., 2012*; *Bolz et al., 2015*; *Danielson et al., 2016*). On the other hand, young cells show enhanced excitability and reduced I/E balance (*Schmidt-Hieber et al., 2004*; *Marín-Burgin et al., 2012*), which was suggested to generate relatively unspecific firing of newborn cells. However, the restricted entorhinal connectivity shown by *Dieni et al. (2016)* together with our new findings, showing efficient burst firing with sparse connectivity, offer a new perspective which might solve the apparent contradiction. At 2 wpi, there are approximately 200-times less glutamatergic synapses (4*50) in young cells. Furthermore, the paired recordings showed that the sparse synaptic connectivity together with the enhanced electrical excitability generates distinct activation profiles of neighbouring young neurons indicating that two adjacent young cells probe for activity in two different non-overlapping sub-populations of afferent fibers. Taken together, this would enable highly specific NMDA-receptor activation by brief burst firing in specific small subsets of afferent fibers. As increasing synapse numbers with development appear to be balanced with decreasing excitability, the synaptic connectivity gradually shifts from a low-number-high-impact state towards a high-number-low-impact scenario.

It is well known that synaptic and extrasynaptic dendritic NMDA receptors are involved in the formation of new synapses during long-term potentiation (*Durand et al., 1996*; *Engert and Bonhoeffer, 1999*; *Maletic-Savatic et al., 1999*). It was shown in cortical pyramidal cells of developing rats that a small number of brief glutamate pulses onto a dendritic shaft is sufficient to induce rapid growth of new spines, dependent on the activation of extrasynaptic NMDA receptors (*Kwon and Sabatini, 2011*). Newly generated granule cells express a high density of extrasynaptic NMDA receptors during the time period of synapse formation (*Schmidt-Salzmann et al., 2014*). Furthermore, it was suggested that newly generated granule cells form new synapses with entorhinal boutons previously synapsing onto spines of preexisting mature granule cells in a competitive manner (*Toni et al., 2007*; *Adlaf et al., 2017*). Together with our new data, this would suggest that NMDA-receptor activation and AP firing in young neurons by coincident brief burst firing support new synapse formation within a restricted number of non-overlapping sub-populations of afferent fibers favouring sparse coding of neuronal input information.

At early developmental stages (<4 weeks post mitosis) there is only limited information about output synapses. However, at 14 dpi the young granule cells form already an axon with ~1000 μm

length and synapses in CA3 (*Faulkner et al., 2008*). At 17 dpi large mossy fibre boutons (MFB) in CA3 were reported, and at 21 dpi the axon length and MFB density are already close to mature values indicating that the neurons are able to contribute to hippocampal network processing (*Toni et al., 2008*; *Sun et al., 2013*). As a consequence, the young neurons would be able to contribute to learning and memory as soon as 3–4 weeks after mitosis in a unique and input specific manner as suggested by behavioral data (*Nakashiba et al., 2012*; *Gu et al., 2012*).

Taken together, our data show for the first time NMDA-receptor dependent AP-firing in young hippocampal neurons, evoked by glutamatergic synapses largely devoid of AMPA receptors. We have identified several key factors including the high input resistance, the slow membrane time constant and the slow gating kinetics of NMDA receptors, which provide the basis for dynamic NMDA-receptor unblocking and AP firing driven by a small number of 'silent' synapses. Finally, the enhanced synaptic excitability does not generate unspecific firing in young neurons. By contrast, it allows for a more restricted activation of the young neurons with small non-overlapping sparse connectivity, well suited to support hippocampal information processing.

## Materials and methods

### Slice preparation

Heterozygous transgenic mice expressing the red fluorescent protein DsRed under the control of the doublecortin (DCX) promoter were maintained in a C57BL/6 background (*Couillard-Despres et al., 2006*). At 6–8 weeks of age, the mice were housed in groups of 4–10 animals at a 12:12 light-dark cycle in large cages with running wheels and enriched environment for at least 2 weeks prior to experiments. Prior to decapitation, mice were firstly kept in an oxygen-enriched environment for 10–15 min, and subsequently anaesthetized with isoflurane (4% in $O_2$, Vapor, Draeger). All procedures are in accordance with national and institutional guidelines. Transverse 300- to 350-μm-thick hippocampal brain slices were cut using a Leica VT1200 vibratome (*Geiger et al., 2002*; *Bischofberger et al., 2006*). For cutting and storage, a sucrose-based solution was used, containing 87 NaCl, 25 $NaHCO_3$, 2.5 KCl, 1.25 $NaH_2PO_4$, 75 sucrose, 0.5 $CaCl_2$, 7 $MgCl_2$ and 10 glucose (equilibrated with 95% $O_2$/5% $CO_2$). Slices were kept at 35°C for 30 min after slicing and subsequently stored at room temperature until experiments were performed.

### Stereotaxic viral injections

In 2 month old male C57BL/6 mice dividing neuronal progenitor cells and their progeny were labeled using a Moloney murine leukemia virus containing a green fluorescent protein (GFP) expression cassette under the control of the CAG promoter (*Zhao et al., 2006*). Viral injections were performed as previously described (*Sultan et al., 2013*; *2015*). Mice were anesthetized with 4% isoflurane and then maintained at a surgical plane by continuous inhalation of 2% isoflurane. They were placed in a stereotaxic frame (Narishige Scientific Instruments, Tokyo, Japan) and 1.5 μl of virus at a titer of $10^7$–$10^8$ pfu ml$^{-1}$ was injected into the dentate gyrus at the following coordinates from the Bregma: –2 mm antero-posterior, 1.75 mm lateral and –2.00 mm dorso-ventral, using a calibrated 5 μl Hamilton syringe fitted with a 33-gauge needle.

### Histochemistry

Histochemical analysis of GCs was performed as described previously (*Couillard-Despres et al., 2006*). Cells were filled with biocytin (2 mg ml$^{-1}$) during whole-cell recording (see below) and stored afterwards at room temperature for at least 1 hr. Acute brain slices were fixed overnight in 4% paraformaldehyde and then incubated for 24 hr with 0.3% triton X-100 (Sigma-Aldrich) together with FITC-conjugated avidin-D (2 μl ml$^{-1}$, Vector, RRID:AB_2336455) at 4°C to visualize biocytin filling. After washing, the slices were embedded in Prolong Gold (Molecular Probes). Fluorescence labeling was analyzed with a confocal laser scanning microscope (LSM 700, Zeiss) using a 40x oil-immersion objective (NA 1.4) acquiring 1 μm-thick optical sections.

The morphology of the dendritic trees of biocytin-filled young and mature granule cells was analysed after generating maximum intensity projections from confocal stacks covering the full size of neurons using ImageJ. The diameter of the dendritic cone was measured parallel to the GCL, either at the base corresponding to the widest part (dendritic cone diameter) or at 100 μm distance from

the soma (diameter at 100 µm). The maximal extension of the dendritic tree corresponding to the height of the cone was measured from the soma towards cone base in approximately radial direction from the GCL.

## Electrophysiology

Electrophysiological experiments were performed using an upright microscope (Examiner.D1, Zeiss, Oberkochen, Germany), equipped with a confocal laser scanning head (LSM 700, Zeiss). Mature granule cells were identified in the dentate gyrus by the location of the cell soma at the outer border of the granule cell layer close to the molecular layer, using infrared-differential-interference-contrast (IR-DIC) video-microscopy. Additionally, the input resistance was measured and confirmed to be smaller than $R_{in} < 400$ MΩ as previously described (*Schmidt-Hieber et al., 2004*; *Heigele et al., 2016*). Newly generated young granule cells (young GCs) were identified in the inner border of the granule cell layer by simultaneous confocal detection of DsRed or GFP fluorescence. In DCX-DsRed transgenic mice, young and mature GCs were recorded in a randomly selected section of the dentate gyrus. Radially oriented fluorescent cells with their soma located in the granule cell layer were selected. Horizontally oriented fluorescent cells in the subgranular zone were not analyzed. During the electrophysiological experiments, slices were continuously superfused with artificial cerebrospinal fluid (ACSF) containing (in mm): 125 NaCl, 25 NaHCO$_3$, 25 glucose, 3 KCl, 1 NaH$_2$PO$_4$, 2 CaCl$_2$, 1 MgCl$_2$ (equilibrated with 95% O$_2$/5% CO$_2$).

Glutamatergic synaptic currents and potentials were isolated by pharmacological block of GABAergic currents using 100 µM picrotoxin. 10 µM CNQX was used to block AMPA receptor mediated currents and 25 µM AP5 (D-(-)-2-amino-5-phosphonopentanoic acid) to block NMDA-receptor mediated currents as indicated.

For recordings of newborn young granule cells, patch pipettes (7–12 MΩ) were pulled from borosilicate glass tubing with 2.0 mm outer diameter and 0.7 mm wall thickness (Hilgenberg, Malsfeld, Germany; P-2000 laser puller, Sutter Instruments, Novato, USA), gently fire-polished and filled with an intracellular solution containing (in mM): 135 K-gluconate, 21 KCl, 2 MgCl$_2$, 2 Na$_2$ATP, 0.3 NaGTP, 10 HEPES, 10 EGTA (adjusted to pH 7.3 with KOH). For voltage-clamp measurements of NMDA- and AMPA-currents pipette solution was (in mM): 100 Cs-gluconate, 20 CsCl, 2 TEACl, 2 MgCl$_2$, 2 Na$_2$ATP, 0.3 NaGTP, 10 Na-Phosphocreatine, 10 HEPES, 10 EGTA (adjusted to pH 7.3 with CsOH).

Voltage signals and currents were measured with a Multiclamp 700B amplifier (Molecular Devices, Palo Alto, CA, USA), filtered at 10 kHz for current and voltage clamp, and then digitized with 20 kHz using a 16-bit CED Power 1401 mk II interface (Cambridge Electronic Design, Cambridge, UK). Bridge balance was used to fully compensate the series resistance ($R_S$) in current clamp recordings ($R_S \approx$ 10–50 MΩ). In voltage clamp, series resistance was compensated by ~80% and experiments were discarded if $R_S$ changed by more than 20% during the recordings. Data acquisition was achieved using IGOR Pro 6.31 (WaveMetrics, Lake Oswego, Oregon) and the CFS library support from CED (Cambridge Electronic Design, Cambridge, UK). Data analysis was performed using the custom made open source analysis software Stimfit (https://neurodroid.github.io/stimfit, *Guzman et al., 2014*) and Mathematica 10 (Wolfram Research, US). As newborn DCX-positive granule cells show a remarkably high input resistance (typical range 1.5–10 GΩ), the pipette seal resistance ($R_{Seal}$) was carefully monitored (*Heigele et al., 2016*). On average the seal resistance was 40.1 ± 2.1 GΩ ($n$ = 93) corresponding to 12.1 ± 0.9 times $R_{in}$. All recordings were performed at 25 ± 2°C. All chemicals were obtained from Tocris, Sigma or Merck.

## Extracellular synaptic stimulation

For stimulation of presynaptic fibers, 4–6 MΩ pipettes filled with HEPES-buffered Na$^+$-rich solution were used to apply brief negative current pulses (10–50 µA, 200 µs). Pipettes were placed in the middle third of the molecular layer (ML), within approximately 100 µm laterally to the recorded cell, if not stated otherwise. For all experiments except the ones in ACSF, the optimal position of the stimulation electrode was searched before neurons were recorded in whole-cell patch-clamp configuration. Therefore, a brief burst of 5 stimuli was generated at 50 Hz in the presence of picrotoxin every 30 s, and the position of the stimulation electrode was varied until a synaptically evoked action potential could be recorded in cell-attached configuration. This position approximately

corresponded to the center of the dendritic firing field also leading to largest current responses in whole-cell voltage-clamp configuration. Subthreshold EPSCs and EPSPs were studied by repeating stimulation protocols 5 times at a frequency of 0.03 Hz and averaged sweeps were analyzed.

## Data analysis

### Intrinsic cell properties

Intrinsic properties were determined within the first minutes in whole-cell configuration. Membrane potentials were measured without correction for liquid junction potentials. Resting membrane potentials were measured in the I = 0 mode. Only cells were included in further analysis with resting $V_m$ more negative than −60 mV (**Heigele et al., 2016**). The input resistance and the membrane capacitance were determined in voltage clamp from the current response to a negative voltage pulse (–5 mV, 500 ms) from a holding potential of –80 mV. The membrane time constant ($\tau_m$) was estimated in current-clamp mode by fitting a monoexponential function to the voltage decay after a small 1-s-current-pulse leading to approximately 5 mV hyperpolarization. Stimulation artifacts were blanked in all voltage clamp figures.

### Estimation of cell age from Rin

Transgenic animals expressing the red fluorescent protein DsRed under the control of the double-cortin (DCX) promotor can be used to identify newborn granule cells younger than 4 weeks post mitosis (**Couillard-Despres et al., 2006**; **Heigele et al., 2016**). To approximately estimate the post-mitotic age of DCX-DsRed cells in the adult hippocampus, we firstly analyzed the decay of $R_{in}$ after mitosis using retrovirus-GFP-labelled neurons (**Figure 1—figure supplement 1c**) and then used this calibration curve to estimate the age of DCX-DsRed neurons from $R_{in}$. To obtain the calibration curve, the $R_{in}$ was measured in young granule cells at a time $t$ of 7 ± 1 dpi (32 ± 9 GΩ, n = 4), 14 ± 1 dpi (5.9 ± 0.5 GΩ, n = 41), 21 ± 1 dpi (1.8 ± 0.5 GΩ, n = 16), 28 ± 1 dpi (0.33 ± 0.04 GΩ, n = 13) and 42 ± 1 dpi (0.37 ± 0.11 GΩ, n = 3). Furthermore, the $R_{in}$ of mature neurons located at the outer border of the granule cell layer in 2–3 month old mice was measured ($R_{in}$ = 0.21 ± 0.01 GΩ, n = 56) and included in the fit with an assumed age of 10 weeks. Fitting the exponential function:

$$R_{in}(t) = \mathrm{R}_\infty + (\mathrm{R}_0 - \mathrm{R}_\infty)\exp(-t/\tau) \tag{1}$$

to the data resulted in the following fitted parameters: $R_\infty$=0.284 GΩ, $R_0$ = 180 GΩ and $\tau$ = 4.04 days consistent with an about 4.5-fold decay of $R_{in}$ per week similar to previous estimates (**Heigele et al., 2016**).

To validate this procedure and to test whether such a fitted function would be useful to calculate cell age, we back-calculated the age of a cohort of virus-GFP-labeled neurons (7–33 dpi) and compared the estimated age with the actual age after virus injection by linear regression analysis (**Figure 1—figure supplement 1d**). Finally, we used an estimated age of 13–18 days (corresponding to 8–2.5 GΩ) to classify cells as 2–3 weeks old and an estimated age of 19–27 days (corresponding to 2–0.5 GΩ) to classify cells as 3- to 4-week-old neurons.

### Data fitting methods

The correlation of the cell firing probability $f(stim)$ with the extracellular stimulation intensities was analyzed by fitting the data with the sigmoidal equation:

$$f(stim) = 1/(1 + \exp(-(stim - \mathrm{EC}_{50})/\mathrm{slope})) \tag{2}$$

with $stim$ referring to the stimulation intensity, $EC_{50}$ to the stimulation intensity inducing 50% firing probability and slope referring to the slope factor.

To describe the variation of firing probability $f(x)$ dependent on tangential location of the stimulation electrode, data were fitted with the bell-shaped function:

$$f(x) = 1/(1 + (\mathrm{abs}((x\text{-}c)/\mathrm{a}))^{2\mathrm{b}}) \tag{3}$$

with $x$ referring to the spatial location of stimulation electrode. The parameter a is the half width of the bell-shaped curve, b/(2a) determines the slope at 50% firing probability and c locates the center of the curve.

The dependence of the amplitude of AMPA- and NMDA-receptor mediated currents ($Amp(R_{in})$) on the input resistance $R_{in}$ (*Figure 2*) was fitted after transforming $R_{in}$ to logarithmic scale which is linearly correlated with maturation time (*Heigele et al., 2016*):

$$Amp\,(R_{in}) = A_0 \exp(-\log(R_{in}/0.2)/\log(R_1/0.2))/(1 + \exp((\log R_{in} - \log R_{50})/\log R_{slope})) \qquad (4)$$

with $A_0$ referring to the average amplitude of mature cells (at 0.2 GΩ). The resistance $R_1$ determines the initial decrease in current amplitude, which decays e-fold when the input resistance increases by $R_1$. An additional sigmoidal decay with increasing $R_{in}$ is shaped by the resistance at half-maximal amplitude $R_{50}$ and the slope factor $R_{slope}$. The AMPA currents initially decreased e-fold with $R_{in}$ increasing by $R_1 = 0.53$ GΩ, with an additional sigmoidal decay to half-maximal values at $R_{50} = 2.87$ GΩ (slope factor 1.27 GΩ). The NMDA currents initially decreased e-fold with increasing $R_{in}$ by $R_1 = 0.66$ GΩ, with an additional sigmoidal decay to half-maximal values at $R_{50} = 5.17$ GΩ (slope factor 1.34 GΩ).

## Action potential properties

Overshooting action potentials were counted in response to burst stimulation of afferent fibers. At least 5–10 burst repetitions with the same stimulation intensity and location were performed and the proportion of successful trials with at least one overshooting AP was calculated as 'AP firing probability'. The AP amplitude and duration at half-maximal amplitude were measured from a baseline at resting potential (about −80 mV). The spike latency was measured as AP peak latency relative to stimulus onset. For *Figure 8—figure supplement 1d–f* all APs per burst within a time window of 200 ms following stimulus onset were counted and the average number of APs per burst was calculated.

## Statistics

Statistical analysis was performed with GraphPad Prism 6 using a two-sided Mann-Whitney U test or a Wilcoxon matched-pairs signed rank test for unpaired and paired data, respectively. The significance level was set to $p < 0.05$. Average data were given as mean ± s.e.m.

No statistical methods were used to predetermine sample sizes, but the used sample sizes are similar to those generally employed in the field. The numbers given refer exclusively to biological replicates, which was equal to number of cells. Data collection and analysis were not performed blind to the conditions of the experiments to promote discovery of potentially unexpected results.

## Acknowledgements

We thank Jan Schulz and Fiona Doetsch for helpful comments on the manuscript, Selma Becherer for histochemical staining and technical assistance and Martine Schwager for mouse genotyping. Supported by Leenaards foundation and Synapsis foundation to NT and SS and the Swiss National Science Foundation (SNSF, Project 31003A_13301).

# Additional information

### Funding

| Funder | Grant reference number | Author |
| --- | --- | --- |
| Schweizerischer Nationalfonds zur Förderung der Wissenschaftlichen Forschung | Project 31003A_13301 | Josef Bischofberger |

The funders had no role in study design, data collection and interpretation, or the decision to submit the work for publication.

### Author contributions

Liyi Li, Data curation, Formal analysis, Investigation, Methodology, Writing—original draft, Writing—review and editing; Sébastien Sultan, Data curation, Methodology, Writing—review and editing;

Stefanie Heigele, Data curation, Formal analysis, Investigation; Charlotte Schmidt-Salzmann, Conceptualization, Methodology, Writing—review and editing; Nicolas Toni, Conceptualization, Funding acquisition, Methodology, Writing—review and editing; Josef Bischofberger, Conceptualization, Resources, Formal analysis, Supervision, Funding acquisition, Validation, Investigation, Visualization, Methodology, Writing—original draft, Project administration, Writing—review and editing

## Author ORCIDs
Josef Bischofberger (iD) http://orcid.org/0000-0002-4006-1663

## Ethics
Animal experimentation: This study was performed in strict accordance with the recommendations in the FELASA Guide for the Care and Use of Laboratory Animals. The protocol for use and care of experimental animals (mice) in this project was approved by the Animal Ethics Advisory committee of the Kanton Basel (2385_26940, 2438_26489, Kantonales Verterinaeramt BS, Switzerland).

## Decision letter and Author response
Decision letter https://doi.org/10.7554/eLife.23612.013
Author response https://doi.org/10.7554/eLife.23612.014

## Additional files

### Supplementary files
• Transparent reporting form
DOI: https://doi.org/10.7554/eLife.23612.012

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
