## [Decision Letter]

Thank you for submitting your article "Silent synapses drive action potential firing in young neurons of the adult hippocampus" for consideration by *eLife*. Your article has been reviewed by three peer reviewers, one of whom is a member of our Board of Reviewing Editors and the evaluation has been overseen by the Reviewing Editor and Eve Marder as the Senior Editor. The following individual involved in review of your submission has agreed to reveal his identity: Shaoyu Ge (Reviewer #3).

The reviewers have discussed the reviews with one another and the Reviewing Editor has drafted this decision to help you prepare a revised submission.

This study addresses a role for NMDA receptors and sparse synapse connectivity in firing of adult born hippocampal granule cells. The topic is timely and important for understanding the mechanism by which young granule cells contribute to hippocampal circuit function. The main findings are novel and should be of general interest. However, the reviewers raise a number of concerns that must be adequately addressed before the paper can be accepted. In particular, revisions are requested to strengthen the present conclusions and to highlight the significance of the new findings more clearly in the context of previous literature. Please note that points 1 and 3 of essential revisions require additional experiments. For point 3, the reviewers have noted that the authors may already have some data (biocytin fills of mature and young granule cells) that are suitable for analysis.

Essential revisions:

1) The authors need to compellingly demonstrate a requirement for NMDARs in spikes elicited in young granule cells in the presence of intact AMPAR-mediated depolarization. To this end, the representative traces for the experiments summarized in Figure 3 should be included along with Figure 3, and the effect of APV on subthreshold depolarization should be quantified. Also, the claimed sufficiency of NMDARs for spiking in young neurons needs clarification, given the small depolarization that remains in the presence of CNQX+APV (Figure 3).

2) The temporal property of spikes is noticeably different between young and mature neurons: in young neurons, the spikes appear after a delay following the stimulus train. One should analyze the spike latency to clarify the temporal differences, and also measure the spike widths to address the possibility of Ca^2+^ spikes. Moreover, one should include detailed information on how the cell firing probability has been determined.

3) With respect to the narrow firing field and the scarce overlap of dendritic arbor to incoming inputs of young granule cells, it will be informative to compare the dendritic tree size between young and mature neurons in tangential and radial directions by including a fluorescent cell fill or by biocytin fill (the authors appear to possess such data already) and subsequent analysis of synapse density. The spatial relationship between the location of the stimulation, synapse density, and the limit of the dendritic arbor that gives rise to the restricted firing response need to be discussed (also see point 4 below).

4) The authors should provide a more accurate and measured discussion of their results in the context of prior results and interpretations. In particular, the main conclusion about input overlap appears to counter the interpretation of Marin-Burgin et al., (2012) but support the interpretation of Dieni et al., (2016), and yet the authors make little effort to reconcile their results with these prior findings. Dieni et al., (2016) is not mentioned, and Marin-Burgin et al., (2012) is only briefly, and inaccurately, described.

5) In Figure 4, the authors attempt to compare the contribution of perforant-path (OML) vs. hilar mossy cell (IML) inputs for evoking the NMDAR-dependent spiking in young DG neurons. However, the use of DCG-IV with the intention of activating presynaptic mGluR on entorhinal axonal projections is problematic if DCG-IV also activates mGluR2 on granule cells to open GIRK channels (Brunner et al., 2013). Therefore, interpretation of the effects of DCG-IV is not as straightforward as presented. Given that the dissociation between OML and IML inputs is not crucial to the main conclusions of the paper, it is recommended that this figure is removed or strengthened.

[Editors' note: further revisions were requested prior to acceptance, as described below.]

Thank you for resubmitting your work entitled "Silent synapses generate sparse and orthogonal action potential firing in adult-born hippocampal granule cells" for further consideration at *eLife*. Your revised article has been favorably evaluated by Eve Marder (Senior editor) and a Reviewing editor.

The manuscript has been improved but there are some remaining issues that need to be addressed before acceptance, as outlined below:

1) One should clearly acknowledge in the text that one cannot rule out the potential contribution of CNQX- and APV-insensitive depolarization (Figure 3, Figure 4) in driving NMDA receptor activation in young neurons.

2) The interpretation of present results will be further strengthened if the results are discussed in light of previous reports that are highly complementary. Dieni et al., 2013 and 2016 showed that immature GCs have higher spiking probability than mature GCs when the stim electrode is near recorded cells but lower spiking probability when the electrode is far from recorded cells. The inverse relationship between EPSC amplitudes/small dendritic trees and intrinsic excitability, and its contribution to maintaining sparse AP firing was a main point of Dieni, 2013. Further, Dieni, 2016 also used a statistical model to demonstrate how the low overlap in afferent fibers to new neurons promotes orthogonalization of output patterns, very similar to experiments in Figure 7. The authors should articulate the main findings of the existing literature.

---

## [Author Response]

Essential revisions:

*1) The authors need to compellingly demonstrate a requirement for NMDARs in spikes elicited in young granule cells in the presence of intact AMPAR-mediated depolarization. To this end, the representative traces for the experiments summarized in Figure 3 should be included along with Figure 3, and the effect of APV on subthreshold depolarization should be quantified. Also, the claimed sufficiency of NMDARs for spiking in young neurons needs clarification, given the small depolarization that remains in the presence of CNQX+APV (Figure 3).*

We agree that showing the effect of AP5 in the presence of intact AMPAR is important. Thus, we have inserted new traces to Figure 4 (former Figure 3) to convincingly show this. The effect of AP5 on subthreshold EPSPs was included already as supplement previously. To strengthen this point we have now moved these data from supplement to a new Figure 3 and quantified the effect in the Results section as requested.

*2) The temporal property of spikes is noticeably different between young and mature neurons: in young neurons, the spikes appear after a delay following the stimulus train. One should analyze the spike latency to clarify the temporal differences, and also measure the spike widths to address the possibility of Ca^2+^ spikes. Moreover, one should include detailed information on how the cell firing probability has been determined.*

Details about quantification of spike firing probability are now in the Materials and methods section. Properties of burst evoked APs were analysed and reported in the Results subsection “Efficient NMDA-dependent spiking in young granule cells “including AP amplitude, AP half-duration and latency as requested. The half-duration of spikes in young cells (5.2 ± 0.8 ms) and the amplitude (109 ± 3 mV) of APs are similar to previously reported TTX-sensitive APs in adult born granule cells. By contrast, spikes were much larger and shorter than TTX-resistant Ca^2+^-spikes in young granule cells (half-duration 87 ± 8ms, amplitude 18 ± 1mV, Schmidt-Hieber et al., 2004), clearly showing that we are not counting Ca^2+^-spikes.

3) With respect to the narrow firing field and the scarce overlap of dendritic arbor to incoming inputs of young granule cells, it will be informative to compare the dendritic tree size between young and mature neurons in tangential and radial directions by including a fluorescent cell fill or by biocytin fill (the authors appear to possess such data already) and subsequent analysis of synapse density. The spatial relationship between the location of the stimulation, synapse density, and the limit of the dendritic arbor that gives rise to the restricted firing response need to be discussed (also see point 4 below).

We would like to thank the referee for this suggestion. We performed morphological analysis of biocytin-filled neurons and report the data in the Results section and in a new Figure 5. This morphological analysis shows a cone-like narrow dendritic arborisation in young cells which is 4-times more narrow than mature and fits nicely to the electro–physiologically determined narrow dendritic firing fields.

*4) The authors should provide a more accurate and measured discussion of their results in the context of prior results and interpretations. In particular, the main conclusion about input overlap appears to counter the interpretation of Marin-Burgin et al., (2012) but support the interpretation of Dieni et al., (2016), and yet the authors make little effort to reconcile their results with these prior findings. Dieni et al., (2016) is not mentioned, and Marin-Burgin et al., (2012) is only briefly, and inaccurately, described.*

We would like to apologize for the inaccurate references. Being more precise, we now mention in the Introduction the low I/E ratio proposed by Marin-Burgin et al., (2012), as well as the low excitatory connectivity proposed by Dieni et al., (2016). Furthermore we discuss the potential implications of our data in the context of Dieni et al., (2016) and Marin-Burgin, (2012) in subsections “AP firing by silent synapses “and subsection “Functional implications for synapse formation and information processing “, respectively.

5) In Figure 4, the authors attempt to compare the contribution of perforant-path (OML) vs. hilar mossy cell (IML) inputs for evoking the NMDAR-dependent spiking in young DG neurons. However, the use of DCG-IV with the intention of activating presynaptic mGluR on entorhinal axonal projections is problematic if DCG-IV also activates mGluR2 on granule cells to open GIRK channels (Brunner et al., 2013). Therefore, interpretation of the effects of DCG-IV is not as straightforward as presented. Given that the dissociation between OML and IML inputs is not crucial to the main conclusions of the paper, it is recommended that this figure is removed or strengthened.

We thank the referees for drawing our attention to the Brunner paper. We believe that DCG4- resistant component in the inner ML might serve as a control, arguing against a strong GIRK channel mediated effect on young GC dendrites. However, dendritic signal processing is complex and a detailed analysis would definitively exceed the scope of the present study.

Therefore, we decided to remove this figure. Nevertheless, we would like to keep a ‘speculative’ sentence in the Discussion section.

[Editors' note: further revisions were requested prior to acceptance, as described below.]

The manuscript has been improved but there are some remaining issues that need to be addressed before acceptance, as outlined below:

1) One should clearly acknowledge in the text that one cannot rule out the potential contribution of CNQX- and APV-insensitive depolarization (Figure 3, Figure 4) in driving NMDA receptor activation in young neurons.

This is now written in Results subsection “Efficient NMDA-dependent spiking in young granule cells”.

2) The interpretation of present results will be further strengthened if the results are discussed in light of previous reports that are highly complementary. Dieni et al., 2013 and 2016 showed that immature GCs have higher spiking probability than mature GCs when the stim electrode is near recorded cells but lower spiking probability when the electrode is far from recorded cells. The inverse relationship between EPSC amplitudes/small dendritic trees and intrinsic excitability, and its contribution to maintaining sparse AP firing was a main point of Dieni, 2013. Further, Dieni, 2016 also used a statistical model to demonstrate how the low overlap in afferent fibers to new neurons promotes orthogonalization of output patterns, very similar to experiments in Figure 7. The authors should articulate the main findings of the existing literature.

Dieni, 2013 and 2016 are now discussed in more detail in subsection “Connectivity with afferent fibers”. Furthermore, an additional reference to Dieni, 2016 is added in subsection “Functional implications for synapse formation and information processing”. To compensate for the additional space we now completely deleted the statements about our old DCG4-experiments.